# Zero-Resource Knowledge-Grounded Dialogue Generation

**Linxiao Li** [*]
Peking University
lilinxiao@pku.edu.cn

**Can Xu**[†]
Microsoft STCA
caxu@microsoft.com

**Wei Wu**
Meituan
wuwei19850318@gmail.com

**Yufan Zhao**
Microsoft STCA
yufzhao@microsoft.com

**Xueliang Zhao**
Peking University
xl.zhao@pku.edu.cn

**Chongyang Tao**
Peking University
chongyangtao@pku.edu.cn

## Abstract

While neural conversation models have shown great potentials towards generating informative and engaging responses via introducing external knowledge, learning such a model often requires knowledge-grounded dialogues that are difficult to obtain. To overcome the data challenge and reduce the cost of building a knowledge-grounded dialogue system, we explore the problem under a zero-resource setting by assuming no context-knowledge-response triples are needed for training. To this end, we propose representing the knowledge that bridges a context and a response and the way that the knowledge is expressed as latent variables, and devise a variational approach that can effectively estimate a generation model from a dialogue corpus and a knowledge corpus that are independent with each other. Evaluation results on three benchmarks of knowledge-grounded dialogue generation indicate that our model can achieve comparable performance with state-of-the-art methods that rely on knowledge-grounded dialogues for training, and exhibits a good generalization ability over different topics and different datasets.

## 1   Introduction

Recent years have witnessed rapid progress on learning a dialogue generation model for open domain human-machine conversation [40, 34, 50, 1]. Though such models in advanced neural architectures [39] are capable of replying with natural and smooth responses regarding to conversation history, people can still feel a clear gap when they converse with the systems, compared with the conversation with humans. One primary reason is that existing dialogue systems lack of necessary knowledge and thus cannot go deep with humans when they dive into a specific topic. To bridge the gap, researchers begin to study how to ground open domain dialogues by external knowledge, which could be obtained either from structured knowledge bases [26, 38], or from unstructured documents [10, 55, 16].

In this work, we study document-grounded dialogue generation in which a response is synthesized regarding to a conversation context associated with a few sentences from external documents. While the documents serve as content sources and hint response generation with knowledge, collecting enough dialogues that are naturally grounded on documents for model training is not trivial. Although some benchmarks built upon crowd-sourcing have been released by recent papers [55, 10, 16], the small training size makes the generation models generalize badly on unseen topics [10] and the cost of building such data also prevents from transferring the technology proved on the benchmarks to

---

[*]Work done during the internship at Microsoft STCA.
[†]Corresponding author: Can Xu (caxu@microsoft.com).

new domains and new languages. A very recent paper [52] attempts to tackle the data challenge under a low-resource assumption, however, reliance on the expensive knowledge-grounded dialogues is still not fully removed. In this paper, we make one step further by exploring knowledge-grounded dialogue generation under a zero-resource setting, where no context-knowledge-response triples (e.g., those obtained from crowd-sourcing) are assumed available in training. Apparently, such an assumption raises even bigger challenges for learning, but our effort will allow developers to build a knowledge-grounded dialogue system from independent dialogues (e.g., context-response pairs collected from Reddit) and knowledge resources (e.g., wiki articles), and thus can greatly reduce the cost of building such systems and enhance transferability of the technology.

Since knowledge-grounded dialogues are absent in training, we introduce two latent variables that represent the knowledge for grounding and the rate of grounding (i.e., how much knowledge is used in responding) respectively. The generation process is then formalized within a probabilistic framework and optimized via variational inference [19]. To take advantage of the recent breakthrough on pre-training for natural language tasks, we build the probabilistic models on the basis of a pre-trained language model. Instead of using generative models, we propose instantiating the posterior with a retrieval model whereby the search space of knowledge is restrained within a few relevant candidates. Thus, we can circumvent the tedious sampling steps and have a more stable learning process. In addition to the objectives in generalized EM, we also devise a knowledge selection loss and a mutual information loss with the former to learn how to tailor long knowledge input to meet the capacity constraint of the pre-trained language model and the latter to effectively estimate the latent grounding rate in variational inference.

We conduct experiments with benchmarks of knowledge-grounded dialogue generation that are constructed by crowd-sourcing. Evaluation results in terms of both automatic metrics and human judgment indicate that our model not only achieves comparable performance with the state-of-the-art model that is learned from crowd-sourced training sets, but also exhibits a good generalization ability over different topics and different datasets.

Our contributions are four-fold: (1) exploration of knowledge-grounded dialogue generation under a zero-resource setting; (2) proposal of a double latent variable model that depicts not only the knowledge connecting a context and a response but also the way that the knowledge is expressed; (3) proposal of a variational learning approach; and (4) empirical verification of the effectiveness of the proposed approach on three benchmarks of knowledge-grounded dialogue generation.

## 2 Approach

Given $\mathcal{D}_{cov} = \{(C_i, R_i)\}_{i=1}^n$ as a dialogue corpus and $\mathcal{K}_{kg} = \{K_j\}_{j=1}^m$ as a knowledge base, where $\forall i \in \{1, \ldots, n\}$, $C_i$ refers to a dialogue context with $R_i$ a response; and $\forall j \in \{1, \ldots, m\}$, $K_j$ denotes a piece of knowledge (e.g., a sentence in Wikipedia), we aim to learn a model $p(R|C, \mathcal{K})$ from $\mathcal{D}_{cov}$ and $\mathcal{K}_{kg}$ without any oracles (e.g., the crowd-workers in existing benchmarks) indicating the collation of a dialogue and the related knowledge. Thus, for a new context $C$ associated with external knowledge $\mathcal{K}$ (e.g., obtained from a retrieval model like in [10]), one can generate a response $R$ following $p(R|C, \mathcal{K})$.

### 2.1 Zero-Resource Learning Framework

Figure 1 gives the graphical model of our approach. The model depicts dependency among four variables: dialogue context $C$, response $R$, latent knowledge $Z_k$, and grounding rate $Z_\alpha$, where $Z_k$ bridges $C$ and $R$ controlled by $Z_\alpha$. Basically, $Z_\alpha$ indicates how much knowledge in $Z_k$ is carried by $R$ according to $C$. Hence, the variable endows our method with flexibility that responses in various levels of knowledge (e.g., from a short reply that simply catches up with the context to an informative statement that delivers necessary content for continuing the discussion) can be modeled in a unified framework. More advantages credited to $Z_\alpha$ include (1) in training, the model guarded by $Z_\alpha$ be-

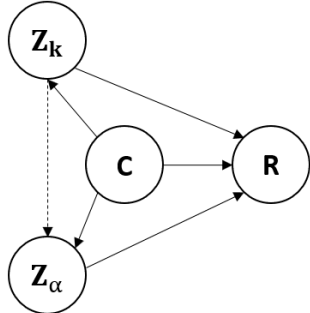

Figure 1: Graphical model of the proposed approach. Solid lines mean that there exists links in both the probabilistic graph and the neural graph, while dotted lines mean that links only exist in the neural graph.

comes more robust regarding to the noise in the inferred $Z_k$; and (2) in prediction, the model can automatically control the way of knowledge expression and thus can be easily adapted to different scenarios without much extra effort. The general objective of learning can be formulated as

$$\mathcal{L}(\theta) = \mathbb{E}_{(C,R)\sim\mathcal{D}_{cov}}[\log p_\theta(R|C)]. \tag{1}$$

By approximating the true posterior with a variational posterior $q(Z_k, Z_\alpha|C, R)$, we optimize the marginal log-likelihood in Eq. 1 with Generalized EM method [4]:

**E-step:**

$$\arg\min_q D_{\mathrm{KL}}(q(Z_k)\|p(Z_k|C,R)) + D_{\mathrm{KL}}(q(Z_\alpha)\|p(Z_\alpha|C,R)), \tag{2}$$

**M-step:**

$$\begin{aligned}
&\arg\max_p \mathbb{E}_{Z_\alpha\sim q(Z_\alpha)}\mathbb{E}_{Z_k\sim q(Z_k)}\log p(R|C,Z_k,Z_\alpha) \\
&- D_{\mathrm{KL}}(q(Z_k)\|p(Z_k|C)) - D_{\mathrm{KL}}(q(Z_\alpha)\|p(Z_\alpha|C,Z_k)),
\end{aligned} \tag{3}$$

where $q(Z_k)$, $q(Z_\alpha)$, and $q(Z_k, Z_\alpha)$ stand for $q(Z_k|C,R)$, $q(Z_\alpha|C,R)$, and $q(Z_k, Z_\alpha|C, R)$, respectively, and $D_{\mathrm{KL}}(\cdot\|\cdot)$ refers to Kullback–Leibler divergence. Detailed derivations are presented in supplementary material.

## 2.2 Neural Parameterization

$\boldsymbol{q(Z_k)}$ **&** $\boldsymbol{p(Z_k|C,R)}$**:** normally, $q(Z_k)$ and $p(Z_k|C,R)$ can be specified as neural generative models (e.g., within the VAE framework [9, 47]). However, learning generative posteriors often requires sampling from a large space that is slow and inaccurate. It is also difficult to approximate the intractable $p(Z_k|C,R)$, which could enlarge the gap between E-step and M-step. Motivated by the issues, we instead define $q(Z_k)$ with a retrieval model. Formally, $q(Z_k)$ is calculated as

$$q(Z_k|C,R) = \frac{exp^{\mathcal{F}(C,R,Z_k)}}{\sum_{K'\in\mathcal{S}(R)} exp^{\mathcal{F}(C,R,K')}}, \tag{4}$$

where $\mathcal{S}(R)$ denotes the inference of the latent knowledge that is made up of top-$l$ results retrieved from $\mathcal{K}_{kg}$ by a relevance model $rel(\cdot,\cdot)$ with $R$ as a query, and $\mathcal{F}(\cdot,\cdot,\cdot)$ is a 3-layer transformer that maps $(C, R, Z_k)$ to a matching score. Since $\mathcal{S}(R)$ is enumerable, $p(Z_k|C,R)$ in Eq. 2 can be calculated by

$$p(Z_k|C,R) = \frac{p(Z_k,R|C)}{p(R|C)} = \frac{p(Z_k|C)p(R|C,Z_k)}{\sum_{K'\in\mathcal{S}(R)} p(K'|C)p(R|C,K')}, \tag{5}$$

where $p(Z_k|C)$ and $p(R|C,Z_k)$ will be detailed later.

$\boldsymbol{p(R|C,Z_k,Z_\alpha)}$**:** we adopt UNILM [11] as the backbone of $P(R|C,Z_k,Z_\alpha)$. Note that UNILM can be replaced by other pre-trained language models such as GPT-2 [28]. Here, we mix $Z_k$ with random noise $\mathcal{Z}$ sampled from $\mathcal{K}_{kg}$. This is to simulate the real scenario in test where the useful knowledge is often enclosed with a lot of irrelevant candidates (e.g., in Wizard of Wikipedia [10], each context is associated with 61 knowledge candidates and only one of them is selected by the crowd-worker for responding). Then $P(R|C,Z_k,Z_\alpha)$ is defined by UNILM($\mathcal{I}$) with $\mathcal{I}$ given by

$$\mathcal{I} = [\text{CLS}][Z_\alpha]\, c_1 \ldots c_{l_c}\, [\text{SEP}]\, S_1 \ldots S_{l_k}\, [\text{SEP}]\, r_1 \ldots r_{l_r}\, [\text{SEP}], \tag{6}$$

where $(c_1, \ldots, c_{l_c})$ denotes the utterance sequence in context $C$, $(r_1, \ldots, r_{l_r})$ denotes the word sequence in response $R$, and $(S_1, \ldots, S_{l_k})$ denotes the sentence sequence of $Z_k \cup \mathcal{Z}$.

One practical issue is that large-scale pre-trained language models such as UNILM often set constraint on the maximum number of tokens they can handle (e.g., 512 tokens in UNILM), which forces us to shorten $\mathcal{I}$ before feeding it to the models. To this end, we devise a knowledge selection model which is formalized as a binary classifier $p(y|C,Z)$ with CLS(UNILM($\mathcal{I}'$)) as input, where CLS($\cdot$) returns the vector corresponding to the $[\text{CLS}]$ token, and $\mathcal{I}' = [\text{CLS}]c_1 \ldots c_{l_c}[\text{SEP}]z_1 \ldots z_{l_z}[\text{SEP}]$ with $Z = (z_1, \ldots, z_{l_z})$. Then, sentences in $Z_k \cup \mathcal{Z}$ are fed to $\mathcal{I}$ one-by-one in a descending order

according to $p(y|C,Z)$ until the capacity constraint. For simplicity, we define $p(Z_k|C)$ in Eq. 5 as $p(y = 1|C, Z_k)$, and define $p(R|C, Z_k)$ in Eq. 5 with $P(R|C, Z_k, Z_\alpha)$ by dropping $[Z_\alpha]$ in Eq. 6.

$p(Z_\alpha|C, Z_k)$: we define $p(Z_\alpha|C, Z_k)$ as $\sigma(\text{CLS}(\mathcal{F}(\mathcal{I}_{p_{Z_\alpha}})))$, where $\sigma(\cdot)$ is a sigmoid function, $\mathcal{F}(\cdot)$ is 3-layer transformer, and $\mathcal{I}_{p_{Z_\alpha}} = e[\text{CLS}]e[Z_\alpha]e[c_1]\dots\ e[c_n]e[\text{SEP}]e[S_1]\dots e[S_k]$ with $e[\cdot]$ the summation of the token embedding, the position embedding, and the segment embedding given by the embedding layer of $\text{UNILM}(\mathcal{I})$.

$q(Z_\alpha)$: $q(Z_\alpha)$ is specified as $\text{Sim}(R, Z_k)$ with $\text{Sim}(\cdot, \cdot)$ a similarity function of sentence pairs.

### 2.3 Learning Details

Besides Eq. 2 and Eq. 3, two extra objectives are also included in learning in order to explicitly optimize knowledge selection and enhance the learning of $Z_\alpha$.

**Knowledge Selection Loss:** the knowledge selection model $p(y|C, Z)$ is optimized by differentiating $Z_{pos}$ from $Z_{neg}$, where $Z_{pos}$ corresponds to the maximum $\text{Sim}(R, Z)$ with $Z \in \mathcal{S}(R)$, and $Z_{neg}$ is randomly sampled from $\mathcal{K}_{kg}$. The loss function can be formulated as

$$\mathcal{L}_{ks} = -\log(p(y = 1|C, Z_{pos})) - \log(p(y = 0|C, Z_{neg})). \tag{7}$$

**Mutual Information Loss:** although the posterior $q(Z_\alpha)$ is deterministic, we still observe that it is hard to encode the information of knowledge expression into $Z_\alpha$ through learning, which is a phenomenon similar to the posterior collapse problem in [5, 51]. To mitigate the problem, we directly impose the association of $Z_\alpha$ and $R$ given $Z_k$ by a mutual information loss defined as $I(Z_\alpha, R) = \mathbb{E}_{p(Z_\alpha, R)} \log \frac{p(Z_\alpha, R)}{p(Z_\alpha)p(R)}$. Since direct optimization of $I(Z_\alpha, R)$ is intractable, we instead propose maximizing a lower bound via variational information maximization [6] which can be formulated as

$$I(Z_\alpha, R) \geq \mathbb{E}_{p(Z_\alpha)}\mathbb{E}_{p(R|Z_\alpha)} \log q_\phi(Z_\alpha|R). \tag{8}$$

In order to optimize Eq. 8, we need to make generation of response tokens differentiable. Recall that the probability distribution of token $w_t$ is calculated as:

$$\mathbf{p}_t = (\mathbf{W}[\mathbf{H}_t]) \in \mathbb{R}^v, \tag{9}$$

where $H_t$ is the hidden state of $w_t$ in $p(R|C, Z_k, Z_\alpha)$, and $\mathbf{W} \in \mathbb{R}^{h \times v}$ are trainable parameters with $h$ the size of $H_t$ and $v$ the vocabulary size. Though one can estimate the gradient of $\mathbb{E}_{p(R|Z_\alpha)}$ with REINFORCE algorithm [43], such an approach often suffers from high variance. Therefore, we instead exploit the gumbel-softmax reparametrization trick [17] as a low-variance approximation of sampling from the categorical distribution $\mathbf{p}_t$:

$$\tilde{e}_t = \sum_{i=1}^{|V|} \bar{e}(w_i) \, \text{softmax}((\mathbf{p}_t + \xi)/\tau)_i, \tag{10}$$

where $\xi$ is an independent noise sampled from the Gumbel distribution, $\tau$ is the temperature (i.e., a hyper-parameter), and $\bar{e}(w_i)$ is the embedding of $w_i$ in $p(R|C, Z_k, Z_\alpha)$. Although this gradient estimator is biased, we find that it works well in practice. We set $\tau = 0.1$ based on the results on validation and fix the value in all the experiments.

The learning algorithm is summarized in Algorithm 1.

### 2.4 Knowledge-grounded Response Generation Model

After learning from $\mathcal{D}_{cov}$ and $\mathcal{K}_{kg}$, we define the response generation model $p(R|C, \mathcal{K})$ in test as $p(R|C, Z, Z_\alpha)$, where we rank $\mathcal{K} = \{K'_i\}$ according to $\{p(y|C, K'_i)\}$ and fill $Z$ with the ranked sequence until reaching the capacity constraint of UNILM, and $Z_\alpha$ is predicted by $p(Z_\alpha|C, Z)$.

## 3 Experiments

We test the proposed method on benchmarks of knowledge-grounded dialogue generation, including Wizard of Wikipedia (Wizard) [10], Topical-Chat (TC) [16], and CMU Document Grounded Conversations (CMU_DoG) [55].

---

**Algorithm 1** Optimization Algorithm

---
1: **Input:** dialogue corpus $\mathcal{D}_{cov}$, knowledge corpus $\mathcal{K}_{kg}$, pre-trained UNILM, threshold $\lambda$, and maximum step $M$.
2: Construct a relevance model $rel(\cdot, \cdot)$ based on $\mathcal{K}_{kg}$.
3: **for** $m \leftarrow 1$ to $M$ **do**
4:    Sample a mini-batch $(C_i, R_i)$ from $\mathcal{D}_{cov}$ and retrieve $\mathcal{S}(R_i)$ with $rel(\cdot, \cdot)$.
5:    Sample a $t$ from uniform(0, 1).
6:    **if** t $< \lambda$:
7:       Update the parameters of the model based on Eq. 7
8:    **else**:
9:       Estimate $p(Z_k|C, R)$ based on Eq. 5
10:       Update the parameters of the model based on Eq. 2                                         ▷ E-Step.
11:       Sample $Z_k$ from $q(Z_k|C, R)$
12:       Update the parameters of the model based on Eq. 3 and Eq. 8                               ▷ M-Step.
13: **end for**
14: **return** $p(y|C, Z)$, $p(Z_\alpha|C, Z)$ and $p(R|C, Z, Z_\alpha)$.

---

## 3.1 Experimental Setup

**Training Data:** we build the knowledge corpus with a Wikipedia dump,[3] where text is extracted with an open source tool[4] and split into sentences using NLTK.[5] In total, there are $5,972,585$ articles and $77,152,626$ sentences. On average, each sentence contains $27.4$ words. The dialogue corpus is constructed from the Reddit Conversation Corpus cleaned by [12]. We merge the training/validation/test sets in the original data, and extract a subset by the following rules: (1) the length of the response falls in $(10, 50)$; (2) the proportion of unique non-stop words in the response falls in $(0.25, 0.6)$; (3) the proportion of unique words in the response is larger than $0.5$; (4) $\text{Sim}(R, \tilde{K}) \geq 0.1$ where $\tilde{K} = \arg\max_{K \in \mathcal{S}(R)} \text{Sim}(R, K)$; and (5) the length of $\tilde{K}$ in (4) is longer than $10$. These rules could remove responses that are too short, too long, too generic, or in an extreme chat-style, and thus can guarantee the quality of training. Automatic evaluation metrics are also sensitive to the length of generated responses. Our model suffers because of the length inconsistent between training and testing. Instead of adjusting the length distribution of training data, we drop the ending token for short responses during training to approximate the maximum average length of benchmarks(24 in our experiment). After the pre-processing, the subset is randomly split into a training set and a validation set with $842,521$ and $2,737$ dialogues respectively. On average, each dialogue (with the last turn as the response and other turns as the context) contains $3.1$ utterances in both sets, and the average length of the utterances is $16.0$ in training and is $16.1$ in validation. Note that the validation set is used for model selection and thus we do not access any data point in the benchmarks before evaluation.

**Test Data:** all the benchmarks are built with crowd-sourcing on Amazon Mechanical Turk (AMT), and are split into training sets, validation sets, and test sets by the data owners. In Wizard and CMU_DoG, knowledge is obtained from Wikipedia, while in TC, besides wiki articles, Washington Post articles and Reddit fun facts are also utilized as the knowledge sources. Unlike CMU_DoG that focuses on movie domain, both Wizard and TC cover a wide range of topics from multiple domains. Various configurations are set up to simulate conversation scenarios in real world. In Wizard, a wizard tells an apprentice about what he/she learns from the knowledge about a specific topic. In addition to wizard-apprentice conversations, CMU_DoG also contains conversations between two workers who know the background documents and try to discuss the content in depth. In TC, participants play symmetric and asymmetric roles according to the knowledge they can access under $5$ settings. In Wizard and TC, the test sets are further split into Seen/Frequent and Unseen/Rare where the former contains topics frequently appearing in the training sets and the latter contains topics infrequently or never appearing in the training sets. For Wizard, we follow [10] and conduct pre-processing with the code published on ParlAI.[6] For CMU_DoG, we use the version shared at `https://github.com/lizekang/ITDD`. For TC, we utilize the data published in the open source project `https:`

`wizard_of_wikipedia`

`//github.com/alexa/alexa-prize-topical-chat-dataset/`. More details of the benchmarks are shown in supplementary material.

**Baselines:** the following models are selected as baselines: (1) **MTASK-RF** [15]: an early model that also realizes knowledge-grounded conversation without crowd-sourced knowledge-grounded dialogues. To make a fair comparison, we implement the model by strictly following the details in [15], but replace the Twitter data, the Foursquare data, and the Twitter handles used to connect the Twitter conversation and the Foursquare facts with the Reddit data, the Wikipedia data, and an aggregate of the topics in the three benchmarks; (2) **Transformer Memory Network (TMN)** [10]:[6] a transformer architecture augmented by a knowledge memory which is published along with the Wizard data; (3) **Incremental Transformer with Deliberation Decoder (ITDD)** [24]:[7] an encoder-decoder architecture where the encoder incrementally represents multi-turn dialogues and knowledge, and the decoder conducts response decoding in two passes similar to the deliberation network in machine translation; (4) **Sequential Knowledge Transformer (SKT)** [18]:[8] a sequential latent variable model with state-of-the-art performance on knowledge selection. Since human labels that indicate ground-truth knowledge are crucial to the performance of the model and only provided in Wizard data, so we implement SKT with heuristics on Topical-Chat and CMU_DoG (pseudo supervision created by selecting GT-knowledge using Sim(.,.) with the response). (5) **Disentangle Response Decoder (DRD)** [52]: a model that exploits pre-training techniques to tackle the low-resource challenge in knowledge-grounded dialogue generation. We choose the one whose parameters are fine-tuned on the full training data of the benchmarks, as the model exhibits the state-of-the-art performance on Wizard according to [52]. We name our model **ZRKGC**[9], standing for "zero-resource knowledge-grounded conversation" model.

**Evaluation Methods:** following [10], we choose perplexity (PPL) [37] and unigram F1 as the automatic metrics, where F1 is calculated with the code shared at `https://github.com/facebookresearch/ParlAI/blob/master/parlai/core/metrics.py`. Besides, we also examine the performance of the models with human annotations. Since human labor is expensive, manual judgment is applied to Wizard only. Following [52], we randomly sample 500 examples from Test Seen and Test Unseen, and recruit 3 well-educated native speakers as annotators. To each annotator, an example is presented with a context, the associated external knowledge,[10] and model responses (top 1 in beam search) that are randomly shuffled to hide their sources. The annotators then judge the quality of the responses from three aspects, including *fluency*, *context coherence* and *knowledge relevance*, and assign a score in $\{0, 1, 2\}$ (representing "bad", "fair", and "good") to each response for each aspect. Each response receives 3 scores per aspect, and the agreement among the annotators is measured via Fleiss' kappa [14].

### 3.2 Implementation Details

We index the sentences in the knowledge corpus with an open source Lucene.Net,[11] employ the internal ranker of Lucene (basically a BM25 model [31]) as $rel(\cdot, \cdot)$, and set the number of retrieved candidates (i.e., $l$) as 10. The function $\text{Sim}(\cdot, \cdot)$ in Section 2.2 is defined as Bleu-2 [27]. We choose UNILM Base (110M) and implement the model with the code in `https://github.com/microsoft/unilm`. We find that replacing $D_{\text{KL}}(q(Z_\alpha)\|p(Z_\alpha|C, Z_k))$ in Eq. 3 with a mean squared error in optimization can enhance model performance, probably because $Z_\alpha$ is a continuous variable. The model is trained with a batch size 10, a maximum input length 256, and a maximum output length 40. The threshold $\lambda$ and the maximum step $M$ in Algorithm 1 are set as 0.2 and $100,000$ respectively. The learning rate is set as 0.00003 and the warmup step is set as 1000. In training, we evaluate the model per $5,000$ steps on the validation set with unigram F1 [10] as a metric. The training procedure will be terminated if we find F1 begins to drop. To draw a fair comparison, we keep the same evaluation procedure with the existing models. During test time, we exploit beam search with a beam size 5. We apply knowledge selection module $p(y|C, Z)$ to select K knowledge sentences from all M knowledge sentences(M>=K) to meet the capacity constraint of UniLM(e.g., 256 in our setting).

Table 1: Automatic evaluation results.

| Models | Wizard Seen | | Wizard Unseen | | Topical Freq | | Topical Rare | | CMU_DoG | |
|---|---|---|---|---|---|---|---|---|---|---|
| | PPL | F1 | PPL | F1 | PPL | F1 | PPL | F1 | PPL | F1 |
| MTASK-RF [15] | 65.4 | 13.1 | 67.7 | 12.3 | 51.3 | 12.6 | 51.6 | 12.5 | 67.2 | 10.5 |
| TMN [10] | 66.5 | 15.9 | 103.6 | 14.3 | 30.3 | 16.5 | 52.1 | 14.6 | 75.2 | 9.9 |
| ITDD [24] | 17.8 | 16.2 | 44.8 | 11.4 | 21.4 | 15.8 | 24.7 | 14.0 | 26.0 | 10.4 |
| SKT [18] | 52.0 | 19.3 | 81.4 | 16.1 | 25.1 | 17.0 | 35.6 | 14.8 | 41.9 | 9.6 |
| DRD [52] | 19.4 | 19.3 | 23.0 | 17.9 | 25.9 | 14.8 | 28.0 | 15.1 | 54.4 | 10.7 |
| ZRKGC | 40.4 | 18.7 | 41.5 | 18.6 | 44.2 | 16.6 | 42.0 | 16.8 | 53.5 | 12.5 |

Table 2: Human evaluation results.

| Models | Wizard Seen | | | | Wizard Unseen | | | |
|---|---|---|---|---|---|---|---|---|
| | Fluency | Coherence | KG Relevance | Kappa | Fluency | Coherence | KG Relevance | Kappa |
| DRD [52] | 1.72 | 1.65 | 1.12 | 0.62 | 1.60 | 1.57 | 1.14 | 0.66 |
| ZRKGC | 1.79 | 1.73 | 1.16 | 0.61 | 1.71 | 1.70 | 1.18 | 0.69 |

## 3.3 Evaluation Results

Table 1 reports the evaluation results on automatic metrics. In terms of F1, though ZRKGC does not access any training examples in the benchmarks, it still outperforms MTASK-RF, TMN, and ITDD, and achieves a comparable performance with DRD on all the test sets, indicating that the model can effectively learn how to leverage external knowledge feed for response generation through the variational approach. Moreover, unlike the baselines, there is almost no difference for ZRKGC on Test Seen and Test Unseen, which reveals the good generalization ability of the model as an advantage of the zero-resource approach: the model is not influenced by specific training data, and thus performs stably over different topics. We further investigate the generalization ability of ZRKGC by comparing it with DRD trained on different benchmarks. Figure 2 shows the results. Interestingly, when we transfer the DRD model trained on one benchmark to another benchmark, there is always significant performance drop. ZRKGC, on the other hand, is always comparable with the best DRD model on each of the benchmarks, indicating that the model generalizes well not only over different topics but also over different datasets. In other words, DRD may fail in practice due to the discrepancy between training and test, but ZRKGC does not suffer from the issue. ZRKGC is worse than ITDD and DRD in terms of PPL, because PPL is calculated with ground-truth responses in the test sets, and therefore models learned by fitting the same or a similar distribution (e.g., both the training data and the test data are constructed by AMT workers) are more advantageous on the metric. We provide richer results with more metrics in supplementary material.

Table 2 compares ZRKGC with DRD using human judgment. All kapa values exceed $0.6$, indicating substantial agreement among the annotators. We observe that responses from ZRKGC are more fluent and more contextually coherent than those from DRD, thanks to the pre-trained language model. Both models are awkward in terms of properly bringing knowledge into responses, which sheds light on the direction for future effort. Cases for a closer inspection are shown in supplementary material.

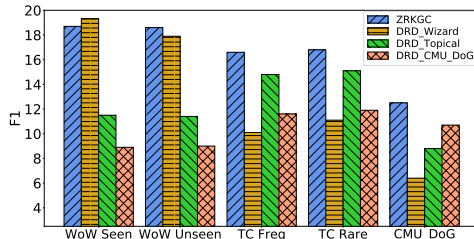

Figure 2: Generalization ability over different datasets.

## 3.4 Discussions

**Retrieval posterior v.s. generative posterior.** We first investigate how the retrieval posterior defined by Eq. 4 matters in learning. To this end, we alternatively implement ZRKGC with a generative posterior (i.e., $q(Z_k)$) that are defined in a sequence-to-sequence form based on UNILM,[12] and check the trajectories of Eq. 3 (i.e., the evidence lower bound (ELBO)) in training under $\{retrieval, generative\} \times \{GEM, ELBO\}$ where GEM means that the model is learned via generalized EM, and ELBO means that optimization is conducted only by Eq. 3. Figure 4 illustrates the trajectories. We can see that with the retrieval posterior, we achieve a tighter ELBO by generalized EM, which means that by optimizing with the E-step, the objective in the M-step moves closer to

Table 3: Ablation study.

| Models | Wizard Seen | | Wizard Unseen | | Topical Freq | | Topical Rare | | CMU_DoG | |
|---|---|---|---|---|---|---|---|---|---|---|
| | PPL | F1 | PPL | F1 | PPL | F1 | PPL | F1 | PPL | F1 |
| ZRKGC | 40.4 | 18.7 | 41.5 | 18.6 | 44.2 | 16.6 | 42.0 | 16.8 | 53.5 | 12.5 |
| $-Z_\alpha$ | 31.1 | 18.5 | 32.0 | 18.4 | 34.2 | 13.9 | 33.2 | 14.4 | 53.2 | 10.8 |
| -mulinfo | 40.9 | 18.1 | 41.9 | 18.0 | 42.6 | 14.2 | 39.1 | 15.2 | 65.7 | 11.7 |
| -retrieval posterior | 35.4 | 16.2 | 36.1 | 16.0 | 39.6 | 13.7 | 36.8 | 14.6 | 52.7 | 10.1 |
| -parameterized posterior | 39.3 | 17.2 | 40.8 | 17.0 | 44.7 | 13.2 | 41.6 | 14.6 | 50.5 | 10.8 |
| -knowledge selection | 44.2 | 18.3 | 45.9 | 17.9 | 45.5 | 14.6 | 43.5 | 14.9 | 53.8 | 12.0 |

the true objective. Since we have to resort to high-variance sampling steps to approximate the KL terms as well as the true posterior (i.e., $p(Z_k|C, R)$) when the generative posterior is used, optimizing with GEM leads to an even worse ELBO than directly executing the M-step. Results in Table 3 also demonstrate that there is a dramatic performance drop (i.e., F1) on the test sets when the retrieval posterior is replaced by a generative posterior (i.e., -retrieval posterior). Moreover, we also observe an obvious drop (i.e., F1) when $\mathcal{S}(R)$ in Eq. 4 is squeezed to $K_{top} = \arg\max_{K \in \mathcal{S}(R)} \mathrm{Sim}(R, K)$ (i.e., -parameterized posterior), indicating the effect of the neural parameterization in Eq. 4.

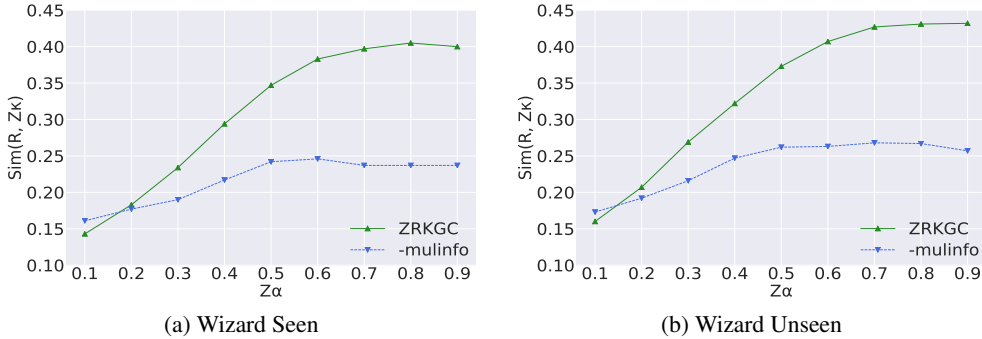

(a) Wizard Seen        (b) Wizard Unseen

Figure 3: Controllability study wrt. knowledge expression.

**Impact of $Z_\alpha$ and impact of the mutual information loss.** Then we study the effect of $Z_\alpha$ in modeling response generation and the effect of the mutual information loss to learning. First, according to the results in Table 3, both removal of $Z_\alpha$ (i.e., ZRKGC becomes a single latent variable model) and removal of the mutual information loss (i.e., -mulinfo) will cause performance drop (i.e., F1), indicating that $Z_\alpha$ is useful to ZRKGC and the mutual information loss can enhance the usefulness of the factor. Recall that $Z_\alpha$ is designed to model knowledge expression and the mutual information loss is designed to effectively learn the factor from data. Thus, we also want

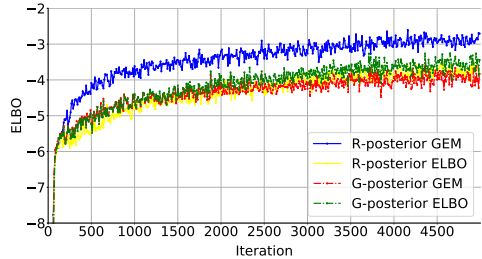

Figure 4: Trajectories of ELBO in training on Wizard.

to check if one can control the extent of knowledge expression by varying $Z_\alpha$ in ZRKGC. Figure 3a and Figure 3b illustrate the comparison between the full ZRKGC and ZRKGC-mulinfo on Test Seen and Test Unseen respectively, in which $Z_\alpha$ is fixed in generation and is increased from $0.1$ to $0.9$ with $0.1$ as the step size, and $\mathrm{Sim}(R, Z_k)$ is employed as the metric with $R$ the generated response and $Z_k$ the ground-truth knowledge.[13] We can see that the gap between the grounding rate of generation and the value of $Z_\alpha$ we set before generation is smaller in the full model than that in the ablated model when $Z_\alpha > 0.2$, indicating that with the mutual information loss, $Z_\alpha$ can effectively encode the information of knowledge expression through the variational learning approach. Note that $Z_\alpha$ becomes weak in ZRKGC when it exceeds $0.5$. This is because data with such grounding rates are sparse in training.

**Impact of the knowledge selection loss.** Finally we explored the role of knowledge selection loss. Our knowledge selection model is mainly to shorten the input sequence of knowledge candidates, while previous work [18] focuses on selecting top-1 knowledge. This obvious difference decided that the performance drop is not significant when replacing knowledge selection module with random selection module according to the results in Table 3.

## 4 Related Work

End-to-end response generation for open domain dialogues is inspired by the successful application of neural sequence-to-sequence models on machine translation [37, 39]. On top of the basic architecture [36, 40], various extensions have been made to tackle the safe response problem [22, 44, 51, 46]; to model dialogue history for multi-turn conversation [33, 35]; to control attributes of responses [45, 53, 48, 41, 32]; and to bias responses to some specific personas [23, 49]. Recently, grounding open domain dialogues by external knowledge is emerging as an important topic in research of human-machine conversation [54, 18, 25, 52]. In this work, we study the problem by reducing the demanding training environment to an extreme where only dialogues and documents as a knowledge base are required. To the best of our knowledge, we are the first who prove that a model learned under such a zero-resource setting can achieve comparable performance on benchmarks with the models learned from the expensive knowledge-grounded dialogues constructed by crowd-sourcing. Unsupervised learning and learning from zero resource have attracted widespread attention in natural language generation tasks. In machine translation, typical methods include pivot-based NMT [13, 29, 7], combination of NMT and SMT [21, 30], creation of pseudo pairs with back translation [2], and adversarial training [20]. In unsupervised abstractive summarization, Wang & Lee [42] exploit adversarial training to make the summary human-readable; Chu & Liu [8] exploit mean of the representations from an auto-encoder for multiple documents to decode a summary; and Baziotis et al. [3] propose a differentiable auto-encoder optimized by re-constructing the input document from the generated summary. Our method is similar to variational back-translation. Instead of directly training a (context,response)-to-knowledge backward generation model, we take the variational posterior of the latent knowledge as the backward model to learn the knowledge-grounded dialogue model. Both SKT[18] and PostKS[25] leverage latent variables for knowledge selection. Besides optimization using generalized EM, our model introduces another variable $Z_\alpha$ to dynamically adapt to candidates in different quality while SKT and PostKS assume there always exists GT-knowledge in their candidates.

## 5 Conclusions

We explore knowledge-grounded dialogue generation under a zero-resource setting by proposing a double latent variable model and a variational learning approach. Evaluation results on benchmarks of the task indicate that our model can achieve comparable performance with state-of-the-art methods and exhibits a superior generation ability over different topics and datasets.

## Broader Impact

Endowing a dialogue system with knowledge is definitely an important step towards human-like conversational AI which has been dreamed by AI researchers for years, especially when such a technology becomes cheaper and more transferable. More importantly, research on knowledge-grounded dialogue generation could fundamentally change the experience of human-machine interaction, as a system will be able to evolve along with the external knowledge base being maintained and updated. This may shed light on the effort on building interfaces that allow people to acquire information in a more natural way (i.e., through conversation), rather than just typing a query in a search box and browsing the blue links. However, we never forget the other side of the coin. Apart from the well-known issues in end-to-end conversation models trained from large naturally-occurring datasets [50], a knowledge base may also be deliberately tailored and bring biased content to dialogues, just like biased content posted by content creators on the Web is promoted by a search engine. To prevent the technology from being abused for disinformation, we look forward to more research effort being paid to fake/biased/offensive content detection, and at the same time, encourage developers to carefully choose the content for building the knowledge base of their dialogue system. After all, good external content can regulate the behavior of a dialogue model in response generation, and help

the model overcome its instinct drawbacks inherited from the malicious or biased content hidden in the large scale dialogues obtained from social media for training.

## Footnotes

[3] `http://wikipedia.c3sl.ufpr.br/enwiki/20191120/`

[4] `https://github.com/attardi/wikiextractor/wiki`

[5] `https://www.nltk.org/`

[6] `https://github.com/facebookresearch/ParlAI/blob/master/projects/`

[7]`https://github.com/lizekang/ITDD`

[8]`https://github.com/bckim92/sequential-knowledge-transformer`

[9]Dataset and codes are publicly available at `https://github.com/nlpxucan/ZRKGC`

[10]For ease of labeling, only the ground-truth knowledge is shown to the annotators in Wizard.

[11]`http://lucenenet.apache.org`

[12] $p(Z_k|C)$ in Eq. 3 is also defined in a generative form.

[13]For the sake of controllability study, we make sure that the ground-truth knowledge annotated by humans is involved in generation.

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
