[Supplementary Material · Supplementary_Material.pdf]

# Supplementary Material of "Zero-Resource Knowledge-Grounded Dialogue Generation"

**Linxiao Li** [*]
Peking University
lilinxiao@pku.edu.cn

**Can Xu**[†]
Microsoft STCA
caxu@microsoft.com

**Wei Wu**
Meituan
wuwei19850318@gmail.com

**Yufan Zhao**
Microsoft STCA
yufzhao@microsoft.com

**Xueliang Zhao**
Peking University
xl.zhao@pku.edu.cn

**Chongyang Tao**
Peking University
chongyangtao@pku.edu.cn

## 1 Derivation of Generalized EM

$$
\begin{aligned}
&\log p(R|C)\\
&= \log \frac{p(Z_k, Z_\alpha, R|C)}{p(Z_k, Z_\alpha|C, R)} \quad \text{(by Bayes' rule)}\\
&= \log p(Z_k, Z_\alpha, R|C) - \log p(Z_k, Z_\alpha|C, R) + \log q(Z_k, Z_\alpha|C, R) - \log q(Z_k, Z_\alpha|C, R)\\
&= \log \frac{p(Z_k, Z_\alpha, R|C)}{q(Z_k, Z_\alpha|C, R)} - \log \frac{p(Z_k, Z_\alpha|C, R)}{q(Z_k, Z_\alpha|C, R)}.
\end{aligned}
\tag{1}
$$

If we multiply $q(Z_k, Z_\alpha|C, R)$ on both sides and integrate $Z_k$ and $Z_\alpha$, then the left part of Eq. 1 can be reformulated as

$$
\begin{aligned}
\log p(R|C) &= \int_{Z_k} \int_{Z_\alpha} q(Z_k, Z_\alpha|C, R) \log p(R|C) dZ_\alpha dZ_k\\
&= \log p(R|C) \int_{Z_k} \int_{Z_\alpha} q(Z_k, Z_\alpha|C, R) dZ_\alpha dZ_k\\
&= \log p(R|C);
\end{aligned}
\tag{2}
$$

and the right part of Eq. 1 can be reformulated as

$$
\begin{aligned}
&\log \frac{p(Z_k, Z_\alpha, R|C)}{q(Z_k, Z_\alpha|C, R)} - \log \frac{p(Z_k, Z_\alpha|C, R)}{q(Z_k, Z_\alpha|C, R)}\\
&= \int_{Z_k} \int_{Z_\alpha} q(Z_k, Z_\alpha) \log \frac{p(Z_k, Z_\alpha, R|C)}{q(Z_k, Z_\alpha)} dZ_\alpha dZ_k - \int_{Z_k} \int_{Z_\alpha} q(Z_k, Z_\alpha) \log \frac{p(Z_k, Z_\alpha|C, R)}{q(Z_k, Z_\alpha)} dZ_\alpha dZ_k\\
&= \text{ELBO} + D_{\text{KL}}(q(Z_k, Z_\alpha) \| p(Z_k, Z_\alpha|C, R)),
\end{aligned}
\tag{3}
$$

where ELBO refers to $\int_{Z_k} \int_{Z_\alpha} q(Z_k, Z_\alpha) \log \frac{p(Z_k, Z_\alpha, R|C)}{q(Z_k, Z_\alpha)} dZ_\alpha dZ_k$. According to the mean-field approximation, $q_(Z_k, Z_\alpha) \approx q_(Z_k) q_(Z_\alpha)$. Hence, ELBO and $D_{\text{KL}}(q(Z_k, Z_\alpha) \| p(Z_k, Z_\alpha|C, R))$ can

---

[*]Work done during the internship at Microsoft STCA.

[†]Corresponding author: Can Xu (caxu@microsoft.com).

be re-written as

$$D_{\mathrm{KL}}(q(Z_k, Z_\alpha)\|p(Z_k, Z_\alpha|C, R))$$

$$\approx \int_{Z_k} \int_{Z_\alpha} q(Z_k)q(Z_\alpha) \log \frac{q(Z_k)q(Z_\alpha)}{p(Z_k, Z_\alpha|C, R)} dZ_\alpha dZ_k$$

$$\approx \int_{Z_k} \int_{Z_\alpha} q(Z_k)q(Z_\alpha) \log \frac{q(Z_k)q(Z_\alpha)}{p(Z_k|C, R)p(Z_\alpha|C, R)} dZ_\alpha dZ_k \quad \text{assume } Z_k \perp\!\!\!\perp Z_\alpha|C, R$$

$$= \int_{Z_k} \int_{Z_\alpha} q(Z_k)q(Z_\alpha)[-\log p(Z_k|C, R) - \log p(Z_\alpha|C, R) + \log q(Z_k) + \log q(Z_\alpha)] dZ_\alpha dZ_k$$

$$= \int_{Z_k} \int_{Z_\alpha} q(Z_k)q(Z_\alpha) \log \frac{q(Z_k)}{p(Z_k|C, R)} dZ_\alpha dZ_k + \int_{Z_\alpha} \int_{Z_k} q(Z_\alpha)q(Z_k) \log \frac{q(Z_\alpha)}{p(Z_\alpha|C, R)} dZ_\alpha dZ_k$$

$$= \int_{Z_k} q(Z_k) \log \frac{q(Z_k)}{p(Z_k|C, R)} dZ_k \int_{Z_\alpha} q(Z_\alpha) dZ_\alpha + \int_{Z_\alpha} q(Z_\alpha) \log \frac{q(Z_\alpha)}{p(Z_\alpha|C, R)} dZ_\alpha \int_{Z_k} q(Z_k) dZ_k$$

$$= D_{\mathrm{KL}}(q(Z_k)\|p(Z_k|C, R)) + D_{\mathrm{KL}}(q(Z_\alpha)\|p(Z_\alpha|C, R));$$

ELBO

$$= \int_{Z_k} \int_{Z_\alpha} q(Z_k, Z_\alpha) \log \frac{p(R|C, Z_k, Z_\alpha)p(Z_\alpha|C, Z_k)p(Z_k|C)}{q(Z_k, Z_\alpha)} dZ_\alpha dZ_k$$

$$\approx \int_{Z_k} \int_{Z_\alpha} q(Z_k)q(Z_\alpha) \log p(R|C, Z_k, Z_\alpha) dZ_\alpha dZ_k + \int_{Z_k} \int_{Z_\alpha} q(Z_k)q(Z_\alpha) \log \frac{p(Z_\alpha|C, Z_k)p(Z_k|C)}{q(Z_k)q(Z_\alpha)} dZ_\alpha dZ_k$$

$$= \mathbb{E}_{Z_\alpha}[\mathbb{E}_{Z_k} \log p(R|C, Z_k, Z_\alpha)] - \int_{Z_k} \int_{Z_\alpha} q(Z_k)q(Z_\alpha) \log[\frac{q(Z_k)}{p(Z_k|C)} + \frac{q(Z_\alpha)}{p(Z_\alpha|C, Z_k)}] dZ_\alpha dZ_k$$

$$\approx \mathbb{E}_{Z_\alpha}[\mathbb{E}_{Z_k} \log p(R|C, Z_k, Z_\alpha)] - D_{\mathrm{KL}}(q(Z_k)\|p(Z_k|C)) - D_{\mathrm{KL}}(q(Z_\alpha)\|p(Z_\alpha|C, Z_k)).$$

$$(4)$$

## 2   More Details of the Benchmarks

Table 1 reports some statistics of the three benchmarks of knowledge-grounded dialogue generation. Note that our model only exploits the test sets for evaluation.

Table 1: Statistics of the benchmarks.

| | Wizard of Wikipedia | | | | CMU_DoG | | | Topic_Chat | | | |
|---|---|---|---|---|---|---|---|---|---|---|---|
| | Train | Valid | Test Seen | Test Unseen | Train | Valid | Test | Train | Valid | Test Freq | Test Rare |
| # dialogues | 18,430 | 1,948 | 965 | 968 | 3,373 | 229 | 619 | 8,628 | 1,078 | 539 | 539 |
| Ave_turns / dialogue | 9.0 | 9.1 | 9.0 | 9.1 | 22.2 | 21.8 | 22.0 | 21.8 | 21.7 | 21.8 | 21.8 |
| Ave_length of utterance | 16.4 | 16.4 | 16.4 | 16.1 | 10.9 | 12.2 | 10.9 | 19.5 | 19.8 | 19.5 | 19.5 |

## 3   Comparison with Pre-trained Language Models

Though ZRKGC exhibits comparable or even better performance in comparison with existing models for knowledge-grounded dialogue generation, one may ask what if we compare ZRKGC with a powerful pre-trained language model. To answer the question, we consider the following two models: (1) **UNILM**$_{finetune}$. We fine-tune the Unilm Base model[3] on response generation and knowledge selection with the full training sets of the benchmarks. Note that in both Wizard and TC, human labels for knowledge selection are provided, while in CMU_Dog, since human labels are absent, we learn knowledge selection by heuristically taking the sentence in knowledge having the largest Bleu-2 score with the response as a positive example and a sentence randomly sampled from the background document as a negative example. This model exploits the same pre-trained language model as ZRKGC, but makes full use of the crowd-sourced training resources; and (2) **DialoGPT** (1). A recent model that attains human-close performance in evaluation. The model follows the

architecture of OpenAI GPT-2, and is trained (either from scratch or from OpenAI GPT-2) with 147M Reddit dialogues (1). We choose the model trained from OpenAI GPT-2 with 345M parameters, as it shows the best performance in the evaluation in (1). The model is implemented based on the code shared at `https://github.com/microsoft/DialoGPT`. According to (1), DialoGPT can reply with commensense knowledge in some cases. Therefore, we apply the model to the benchmarks in a zero-resource setting (i.e., without any fine-tuning with the data of the benchmarks). This is to check if ZRKGC can be simply replaced by DialoGPT if we stick to a zero-resource setting.

Table 2 reports evaluation results on automatic metrics and Table 3 shows human evaluation on Wizard. As expected, if we can prepare some training resources, then fine-tuning a pre-trained language model is the best choice, though such resources are expensive to obtain. On the other hand, if we pursue a cheap yet effective solution to knowledge-grounded dialogue generation, then ZRKGC proved its value since one cannot directly apply a pre-trained language model to the task.

Table 2: Automatic evaluation results.

| Models | Wizard Seen | | Wizard Unseen | | Topical Freq | | Topical Rare | | CMU_DoG | |
|---|---|---|---|---|---|---|---|---|---|---|
| | PPL | F1 | PPL | F1 | PPL | F1 | PPL | F1 | PPL | F1 |
| UNILM$_{finetune}$ | 15.7 | 19.4 | 18.6 | 18.5 | 12.7 | 18.4 | 14.5 | 18.6 | 20.6 | 11.0 |
| DialoGPT(1) | 84.0 | 8.4 | 85.9 | 8.1 | 87.6 | 8.3 | 87.9 | 8.5 | 73.4 | 6.9 |
| DRD (2) | 19.4 | 19.3 | 23.0 | 17.9 | 25.9 | 14.8 | 28.0 | 15.1 | 54.4 | 10.7 |
| ZRKGC | 40.4 | 18.7 | 41.5 | 18.6 | 44.2 | 16.6 | 42.0 | 16.8 | 53.5 | 12.5 |

Table 3: Human evaluation results.

| Models | Wizard Seen | | | | Wizard Unseen | | | |
|---|---|---|---|---|---|---|---|---|
| | Fluency | Coherence | KG Relevance | Kappa | Fluency | Coherence | KG Relevance | Kappa |
| UNILM$_{finetune}$ | 1.83 | 1.80 | 1.31 | 0.66 | 1.76 | 1.75 | 1.29 | 0.66 |
| DialoGPT (1) | 1.80 | 1.78 | 0.82 | 0.69 | 1.75 | 1.73 | 0.80 | 0.65 |
| DRD (2) | 1.72 | 1.65 | 1.12 | 0.62 | 1.60 | 1.57 | 1.14 | 0.66 |
| ZRKGC | 1.79 | 1.73 | 1.16 | 0.61 | 1.71 | 1.70 | 1.18 | 0.69 |

## 4   Case Study

Table 4 and Table 5 present some examples from Wizard Seen and Wizard Unseen respectively. In each case, we show the dialogue context, the knowledge (ground-truth), the human response, and responses from different models. We can see that responses from ZRKGC and DRD are well grounded by the provided knowledge, while responses from MTASK-RF, TMN and ITDD in general lack of informative content. Compared with DRD, ZRKGC is better at leveraging the external knowledge feed in the test phase and replies with more informative and more contextually coherent responses, which demonstrates the potential of the model in practice.

Table 4: Examples from Wizard Seen. A and W refer to Apprentice and Wizard respectively.

---

**Topic: Travel**

---

**Context:**
A: I love to travel! I'm actually taking a cruise in September. What about you?
W: I went on a very cool cruise this past January. Cruises are pleasure voyages and the voyage along with the amenities are all part of the experience. We traveled to Mexico.
A: Oh cool! What part of Mexico? I've been to Cancun, but that's it.
**Knowledge:** There are even "cruises to nowhere" or "nowhere voyages" where the ship makes 2–3 night round trips without any ports of call.

---

**(MTASK-RF)** i love to travel ! i ' m going to a cruise in september !
**(TMN)** it 's a great place to travel to the voyage to the cruise .
**(ITDD)** i 'm not sure , but i do know that they are usually called " nowhere " or " nowhere " .
**(DRD)** it 's a federal republic of north america . it 's a federal republic .
**(ZRKGC)** cancun is a ship in the yucatan peninsula . i ' m not sure if you are thinking of that , but it was an adventure to mexico .
**(Human)** Our cruise was a Walking Dead cruise with one stop in Cozumel. There are some cruises that "cruise to nowhere" and the ship just travels for a few days.

---

**Topic: Reading (process)**

---

**Context:**
W: I have not read for a long time.
A: When was the last time you read a book?
W: Last year. I only read news online since I have no time. What is your favorite book?
A: My favourite book is Wuthering Heights. I also read the news quite often. What is your favourite news to read?
**Knowledge:** Wuthering Heights is Emily Brontë's only novel.

---

**(MTASK-RF)** i have n ' t read a book for a long time . i ' m not a fan of books .
**(TMN)** i love the classics like the truman book .
**(ITDD)** i think it 's a great book . it 's a great book .
**(DRD)** i read the book wuthering heights .
**(ZRKGC)** wuthering heights is a book by emily bronte . it ' s the best story of all time , and i like to read her .
**(Human)** it was written by emily bronte , if i am not mistaken .

---

Table 5: Examples from Wizard Unseen. A and W refer to Apprentice and Wizard respectively.

---

**Topic: Cheerleading**

---

**Context:**
A: I was a cheerleader in high school. Were you ever a cheer leader?
**Knowledge:** Cheerleading is an activity wherein the participants (referred to as "cheerleaders") cheer for their team as a form of encouragement.

---

**(MTASK-RF)** i was a cheerleader in high school . i ' m still a cheerleadingleader .
**(TMN)** i were a big fan of cheerleading , illinois .
**(ITDD)** i 'm not sure , but i 'm not sure about that .
**(DRD)** i was a cheer leader in the middle ages .
**(ZRKGC)** i ' m a cheerleader in the us . it was my first cheerleading camp , and they were so good for me .
**(Human)** Yeah ! Remember back then,we do encourage our college basketball team to success.

---

**Topic: Guns N' Roses**

---

**Context:**
A: Hi. Omg, i REALLY love Guns N' Roses! Best band ever.
W: Yeah, GNR from Los Angeles, California, formed in 1985.
A: I've been their fan for ages. Shadow of your love is my favorite song. What is your favorite song?
W: "Appetite for Destruction" was my favorite album .
A: That's a very good album! I can't believe it came out in 1987!
W: Yeah! Long time ago. Sweet Child o' Mine came out on that album.
A: I totally forgot about that song! Didn't listen to it for a while!
W: You know that album sold 30 million copies!
A: Really? That's insane! I didn't know that .
**Knowledge:** The album has sold approximately 30 million copies worldwide, including 18 million units in the United States, making it the best-selling debut album of all time!

---

**(MTASK-RF)** sweet child o ' mine came out on that album . i love it .
**(TMN)** yeah , i did n't know that . i know that the album was released . it was a great band .
**(ITDD)** it was founded in 2016 .
**(DRD)** yeah , it was released in 2001 by the american film warner bros .
**(ZRKGC)** it ' s the best selling album of all time . i have to be sure that is not a record .
**(Human)** Best selling album of all time n the US!

---

# 5 More Results on Automatic Metrics

Table 6 provides extensive results on metrics other than PPL and F1 in the main paper. Here, BLEU and BOW embedding are calculated with an open-source `https://github.com/Maluuba/nlg-eval`.

Table 6: Extensive results on automatic metrics.

| Method | PPL | F1 | BLEU-1 | BLEU-2 | BLEU-3 | BLEU-4 | Average | Extrema | Greedy |
|---|---|---|---|---|---|---|---|---|---|
| *(WOW-seen)* | | | | | | | | | |
| MTASK-RF | 65.4 | 13.1 | 0.188 | 0.078 | 0.041 | 0.022 | 0.842 | 0.377 | 0.637 |
| TMN | 66.5 | 15.9 | 0.184 | 0.073 | 0.033 | 0.017 | 0.844 | 0.427 | 0.658 |
| ITDD | 17.8 | 16.2 | 0.158 | 0.071 | 0.040 | 0.025 | 0.841 | 0.425 | 0.654 |
| DRD | 19.4 | 19.3 | 0.229 | 0.112 | 0.066 | 0.044 | 0.864 | 0.455 | 0.679 |
| **ZRKGC** | 40.4 | 18.7 | 0.237 | 0.087 | 0.039 | 0.018 | 0.888 | 0.438 | 0.682 |
| *(WOW-unseen)* | | | | | | | | | |
| MTASK-RF | 67.7 | 12.3 | 0.180 | 0.072 | 0.038 | 0.021 | 0.843 | 0.374 | 0.632 |
| TMN | 103.6 | 14.3 | 0.168 | 0.057 | 0.022 | 0.009 | 0.839 | 0.408 | 0.645 |
| ITDD | 44.8 | 11.4 | 0.134 | 0.047 | 0.021 | 0.011 | 0.826 | 0.364 | 0.624 |
| DRD | 23.0 | 17.9 | 0.221 | 0.102 | 0.057 | 0.037 | 0.862 | 0.444 | 0.671 |
| **ZRKGC** | 41.5 | 18.6 | 0.233 | 0.084 | 0.039 | 0.019 | 0.889 | 0.441 | 0.681 |
| *(Topical Freq)* | | | | | | | | | |
| MTASK-RF | 51.3 | 12.6 | 0.182 | 0.077 | 0.043 | 0.025 | 0.879 | 0.403 | 0.655 |
| TMN | 30.3 | 16.5 | 0.176 | 0.079 | 0.041 | 0.025 | 0.891 | 0.444 | 0.693 |
| ITDD | 21.4 | 15.8 | 0.163 | 0.074 | 0.041 | 0.026 | 0.887 | 0.426 | 0.680 |
| DRD | 25.9 | 14.8 | 0.203 | 0.088 | 0.050 | 0.033 | 0.893 | 0.408 | 0.681 |
| **ZRKGC** | 44.2 | 16.6 | 0.231 | 0.083 | 0.039 | 0.021 | 0.890 | 0.431 | 0.680 |
| *(Topical Rare)* | | | | | | | | | |
| MTASK-RF | 51.6 | 12.5 | 0.180 | 0.076 | 0.042 | 0.023 | 0.872 | 0.388 | 0.648 |
| TMN | 52.1 | 14.6 | 0.168 | 0.068 | 0.031 | 0.016 | 0.881 | 0.429 | 0.682 |
| ITDD | 24.7 | 14.0 | 0.153 | 0.062 | 0.032 | 0.019 | 0.880 | 0.408 | 0.670 |
| DRD | 28.0 | 15.1 | 0.190 | 0.083 | 0.046 | 0.030 | 0.874 | 0.398 | 0.667 |
| **ZRKGC** | 42.0 | 16.8 | 0.230 | 0.085 | 0.041 | 0.021 | 0.884 | 0.428 | 0.677 |
| *(CMU_DoG)* | | | | | | | | | |
| MTASK-RF | 67.2 | 10.5 | 0.157 | 0.060 | 0.025 | 0.010 | 0.832 | 0.374 | 0.627 |
| TMN | 75.2 | 9.9 | 0.115 | 0.040 | 0.016 | 0.007 | 0.789 | 0.399 | 0.615 |
| ITDD | 26.0 | 10.4 | 0.095 | 0.036 | 0.017 | 0.009 | 0.748 | 0.390 | 0.587 |
| DRD | 54.4 | 10.7 | 0.150 | 0.057 | 0.025 | 0.012 | 0.809 | 0.413 | 0.633 |
| **ZRKGC** | 53.5 | 12.5 | 0.173 | 0.056 | 0.022 | 0.009 | 0.837 | 0.379 | 0.638 |

## Footnotes

[3]https://unilm.blob.core.windows.net/ckpt/unilm1.2-base-uncased.bin