[Reviews · NeurIPS 2020]

Review 1

Summary and Contributions: Post AR: Thank you for preparing your answers well despite the short author response period. The authors did a good job of addressing many of the concerns of reviewers (especially mine). I believe with these new results, the authors will have a stronger final version. I'm bumping up my score for this reason. ======================================================== The authors propose a novel latent variable model for zero-resource knowledge grounded dialogue. They represent knowledge and its relevance score as latent variables and effectively estimate its values through generalized EM algorithm along with devised knowledge loss and mutual information loss. The proposed model achieves comparable performance on three benchmark datasets, even without fine-tuning.

Strengths: This paper proposes a good direction towards zero-resource knowledge grounded dialogue. They propose a theoretically reasonable learning method along with some efforts to improve its computational efficiency. The experimental setup is also interesting because they measure the model's generalization performance by exploiting several datasets.

Weaknesses: - It is hard to judge whether the proposed method gains good results because of the proposed learning method or the help of the strong pretrained UniLM model. Even though they compare it with DialoGPT in the appendix, I also would like to see the model's performance without UniLM initialization or finetuned DialoGPT with the proposed dataset (e.g., Reddit conversation with top-1 retrieved knowledge). - It is not clear how the proposed model and baselines perform knowledge selection in test time. (i) To the best of my knowledge, ITDD does not have a knowledge selection module. How do you select knowledge for ITDD? (ii) All the examples and details of human evaluation say that authors use ground-truth knowledge. Are all the models use GT knowledge in test time or use top-10 retrieved knowledge from Lucene knowledge retriever? If so, the performance of some baselines would be revised. For example, TMN in Table 1 uses the original WoW setting with an average of 60~70 knowledge sentences in each dialogue turn. (iii) If the authors use the original WoW setting, can you also report knowledge selection accuracy? It would be beneficial for judging the effectiveness of the proposed knowledge selection method. To the best of my knowledge, TMN shows 22.5 with WoW training data and 25.5 with additional Reddit and SQuAD datasets. - Little concern regarding indirect knowledge selection supervision. (i) Number of retrieved knowledge may be too small (10 sentences in this paper) so that all the candidates can be from the same Wikipedia article. Can we say the proposed model learns how to select relevant knowledge in an unsupervised manner? Isn't it just randomly choose from already relevant candidates? What happens if we change # of retrieved knowledge to 32 (or larger or limit the number of sentences from the same article) to weaken the effect of external knowledge retriever? (ii) Table 3 does not contain "without knowledge loss" results. How does the proposed model work without it? (iii) Doesn't the proposed knowledge selection loss (in line #132) make the model just mimic the Lucene knowledge retriever? If I understand correctly, knowledge loss will be 0 if the model completely select same knowledge sentence as the Lucene retriever. - Need to compare with previous works on knowledge-grounded dialogue with latent variables (e.g., Lian et al., "Learning to Select Knowledge for Response Generation in Dialog Systems", IJCAI 2019 and Kim et al., "Sequential Latent Knowledge Selection for Knowledge-Grounded Dialogue", ICLR 2020). To the best of my knowledge, Both papers also set knowledge selection as a latent variable to enable unsupervised or semi-supervised training.

Correctness: Mostly correct but little concern in the knowledge selection supervision part. Please see the section "Weakness" for more details.

Clarity: Pretty well, but missing few (important) details regarding experiments (especially quantitative results part, see section "Weakness" for more details).

Relation to Prior Work: Mostly, but missing some relevant work in using latent variables for knowledge-grounded dialogue. Both below works let knowledge selection as a latent variable in order to infer which knowledge to use in unsuperivsed/semi-supervised manner. I think authors should clearly compare with those works. - Lian et al., "Learning to Select Knowledge for Response Generation in Dialog Systems", IJCAI 2019 - Kim et al., "Sequential Latent Knowledge Selection for Knowledge-Grounded Dialogue", ICLR 2020 - Guu et al., "REALM: Retrieval-Augmented Language Model Pre-Training", ICML 2020 (it is not dialogue task, but pretty related in using latent knowledge-retriever.)

Reproducibility: Yes

Additional Feedback: I appreciate authors for proposing good direction in the area of knowledge-grounded dialogue. My biggest concern is in the details of the experimental setup. I expect authors to organize the experiment more thoroughly during the author response period.


Review 2

Summary and Contributions: 1. The paper proposes a method to learn knowledge-grounded dialog generation without access to context-knowledge-response triples during training. 2. The propose a method to learn latent variables that represent the amount and content of the knowledge for grounding. 3. Comparable results to methods that use knowledge grounded dialogs for training.

Strengths: 1. The problem setting is very relevant and useful in real world. 2. Proposed method shows comparable performance to SOTA methods that do use knowledge-grounded dialogs for training.

Weaknesses: 1. Not clear how the model would scale when the search space for knowledge has to be increased.

Correctness: At a high level, looks correct.

Clarity: 1. Overall, I found the paper hard to read and understand. 2. It would be nice to have an example of the setting (with R, C, K marked ) that the authors care about early on and also an overview of the proposed method in simple words early on. 3. The “Approach” section is tough to understand in some parts. It would be good to have more clarity in that section. 4. It would be useful to have examples of the datasets (In terms of the terminologies used in the “Approach sections”). After author feedback: Thanks for your feedback. Looking at all the other reviews I feel my lack of expertise in the subject area of this paper might have been one main reason why I felt the paper was hard to understand. But still some of the above points might be helpful to improve the paper further.

Relation to Prior Work: Yes

Reproducibility: Yes

Additional Feedback:


Review 3

Summary and Contributions: Post AR: I am still pleased with the paper. W.rt. the response to my question on test time evaluation ("For R3"), I find the response disappointing in that I'm still a little unclear about what happens at deployment time. I also feel like the authors reasonably responded to most of R1's issues. I think the real reason I love this paper is the view of treating UniLM context horizon as a resource constraint in which we need to allocate resources (knowledge). I very much like this view of things, and this is likely going to change my view on several other problems I am working on. The authors consider the task of knowledge-grounded dialogue, where a dialogue agent (chatbot), wishes to have a conversation with a user, while additionally looking up further information to ground on (e.g., reading wikipedia while talking to ensure that responses are factually correct). The authors particularly wish to relax an assumption existing in the literature: that datasets have been collected containing strong annotations marking knowledge relevant to the conversation. Such data is enormously expensive to collect, and so datasets are relatively small in size (Wizard of Wikipedia contains only about 150k utterances, for example). The authors propose a (dual) latent variable to address this problem, in which the retrieved knowledge is assumed to depend on a set of latent, inferred variables. The authors provide a factorization of the model, and an algorithm for optimizing it. The authors evaluate on three existing knowledge-backed datasets, and compare with state-of-the-art models for knowledge-grounded dialogue. Crucially all the existing prior work depends on the strong annotation procedure, while their model does not. Their model outperforms two of the existing models in the literature, and slightly underperforms the best model, despite the lack of supervision. Of particular note is that their models generalize to unseen topics better than strongly-supervised models. The authors provide an ablation of aspects of their model to show that all components are important. The authors also analyze a few nearby variants and show their original proposal is the best.

Strengths: - Relax a serious assumption in the literature about this field of work. This can dramatically lower the cost of collection in the future, or enable rapid adoption to new knowledge bases (e.g. train on wikipedia; deploy on an internal company wiki) - Competitive performance with strongly-supervised models is quite impressive - Paper is thorough in experiments (three datasets, breaks it up by common/uncommon topics, etc; discussion for modeling assumptions like choice of ELBO/gumble softmax, etc) - Hyperparameters and reproduction steps are clear.

Weaknesses: I'm generally happy with the paper. I have a few clarification questions that could be weaknesses.

Correctness: I have no concerns on claims or empirical methodology. Methodology consistent with the literature, and results seem reasonable.

Clarity: No concerns here. The paper is clearly written and does a great job of balancing density for high level explanations (e.g., only critical math is shown in the main text, and is always accompanied by motivations and high level intuitions).

Relation to Prior Work: Yes. They do a good job of comparing to strong models existing in the literature, and they differentiate themselves in a very clear way (don't rely on the strong supervision of grounding).

Reproducibility: Yes

Additional Feedback: The only thing that I don't think I understood reasonably well was on page 3, line 104. The authors say that the K_kg should be ranked by a relevance with R as the query. This seems problematic though: if the KG lookup depends on the response, then I have to know the response ahead of time. While this is okay during training, during deployment I won't be able to do the lookup. What have I misunderstood here, about how this works? (The inclusion of a human evaluation tells me that they DON'T have this issue, I just misunderstand) If I could have one nitpick about the paper, it's that the graphs should be made more readable in black-and-white, but that is obviously unimportant :)


Review 4

Summary and Contributions: The authors present a zero-resource-based approach towards solving knowledge-grounded dialogue. Specifically, the authors propose a model with two latent variables representing 1) grounded knowledge and 2) the rate of usage of the provided knowledge. The authors then train these models with large, pre-existing corpora and evaluate in zero-shot setting on three knowledge-grounded dialogue datasets. Their results indicate that their proposed models generalize much better than models trained on the actual evaluation datasets themselves. In human evaluation studies, their model outperforms the best performing baseline.

Strengths: - The authors propose a cost-effective approach to building knowledge-grounded dialogue agents, in the sense that crowdsourced data is not necessary to build conversational models that can ground their generated responses in knowledge - The authors claim that they are the first to try such a zero-resource approach in the context of knowledge-grounded dialogue generation. - The authors’ approach yields the best human evaluation results in all three metrics (fluency, coherence, knowledge relevance), indicating the strengths of their approach. - The results are relevant to dialogue researchers looking to improve generalizability of knowledge-grounded dialogue models.

Weaknesses: While the authors claim that their approach is “zero-resource,” in reality their evaluations are on datasets that use similar, if not the same, knowledge resources. Thus, it is not entirely clear that their models are truly generalizable - for example, the authors claim that their model performs the best on the wizard unseen test set. However, the authors do not appear to make an effort to remove such topics from their training corpora, which could skew the results. Update post author response: Thank you for your clarification regarding “zero-resource”. Perhaps future work could consider various other KBs and other datasets grounded in those other KBs to further emphasize the generalizability of the proposed method.

Correctness: As noted in the weaknesses section, I am not sure if the authors did due diligence in filtering the training data to ensure that conversations do not discuss any of the topics in the rare/unseen test sets of the CMU_DoG and Wizard datasets respectively.

Clarity: - Section 2.2 on Neural Parameterization would benefit from a clearer approach - many equations are inline and difficult to understand. Additionally, more in-depth definitions of what is represented by various functions (e.g. q(Z_k)) would be beneficial as well. - The paper would benefit from a proofread to correct style and grammatical errors.

Relation to Prior Work: The authors discuss in their related work section how their method differs from previous contributions.

Reproducibility: Yes

Additional Feedback:

[Author Response · NeurIPS 2020]

**For all Reviewers:** Thank you for the valuable comments that help us improve the work.

**For Reviewer #1:** 1. *Effect of UniLM:* We observe obvious performance drop when using fine-tuned UniLM with fixed top-1 retrieved knowledge (-parameterized posterior in Table 3). Following your suggestion, we add an experiment in which we train our model with all parameters randomly initialized, and the results are: $42.9, 18.6, 44.5, 18.5, 44.9, 15.7, 41.3, 16.3, 51.1, 11.8$ (aligned with the columns of Table 1). Therefore, the performance of ZRKGC with UniLM initialization slightly improves but the major improvement comes from our method.

2. *Knowledge selection in the test time:* In case there is any misunderstanding, let us clarify our motivation again. We propose a method to automatically connect Reddit and Wikipedia in replacement of expensive human-annotated dataset (e.g. Wizard) to train a knowledge-grounded generation model. Before us, models are trained on the human-annotated datasets, and are tested with the hold-out data annotated in the same way. Therefore, to draw a fair comparison, we keep the same evaluation procedure with the existing models. Another issue that needs clarification is that in test time the knowledge selection model $p(y|C, Z)$ is responsible for selecting K knowledge sentences from all M knowledge sentences(M>=K) to prevent the length of input from exceeding the maximum length of UniLM while previous work [18][25] focuses on selecting top-1 knowledge. Our model performs implicit knowledge selection on the input K knowledge sentences (concatenated in a sequence) in an end-to-end way like DRD [52]. (i): Yes, for ITDD we do not explicitly select knowledge and simply concatenate all the knowledge candidates into a sequence. (ii): All the models use the original WoW setting with an average of 60~70 knowledge sentences in each dialogue turn. In both human evaluation and case study, all models generate responses with all knowledge sentences. We do not list all knowledge sentences in Table 4 and Table 5, because that will make them hard to read; (iii): For our model, the proportion of the GT-knowledge in the input K knowledge sentences on WoW seen and WoW unseen are $37.7\%$ and $37.4\%$ respectively.

3. *Supervision of knowledge selection:* (i): We have tried varying the number of retrieved knowledge in [1,5,10,15,20,25,30] before. F1 on the validation set increases until the number of knowledge reaches 10, but stays stable when the number increases from 10 to 30. Finally, to speed up training, we use the number 10. We train our model by randomly choose $Z_k$ from the top 10 retrieved knowledge from the Lucene retriever, and the results are: $43.0, 16.7, 44.8, 16.5, 44.6, 12.2, 42.1, 12.9, 53.0, 9.9$ (aligned with the columns of Table 3). Therefore, both randomly sampling and keeping only top-1 knowledge (-parameterized posterior in Table 3) will cause dramatic performance drop, indicating the effectiveness of our method; (ii): The results of "without knowledge loss" are: $44.2, 18.3, 45.9, 17.9, 45.5, 14.6, 43.5, 14.9, 53.8, 12.0$ (aligned with the columns of Table 3). In test time, knowledge selection module is mainly to shorten the input sequence of knowledge candidates, so the performance drop is not significant. (iii): Your understanding is correct. But in practice the Lucene retriever selects knowledge based on responses while the model selects knowledge with only access to contexts and knowledge candidates, so it is very hard for the model to select the same knowledge sentence with Lucene retriever.

4. *Choice of baselines:* We choose DRD instead of SKT [18] and PostKS [25] because (i) human annotations on knowledge selection are crucial to the performance of SKT and PostKS and such annotations (in sentence level) are not available in Topical-Chat and CMU_DoG; and (ii) both SKT and PostKS perform worse than DRD on Wizard [18][25][52]. In spite of this, for your reference, we implement SKT with heuristics on Topical-Chat and CMU_DoG (pseudo supervision created by selecting GT-knowledge using Sim(.,.) with the response), and the results are: $52.0, 19.3, 81.4, 16.1, 25.1, 17.0, 35.6, 14.8, 41.9, 9.6$ (aligned with the columns of Table 1), which are worse than our method. Besides optimization using generalized EM, our model introduces another variable $Z_\alpha$ to dynamically adapt to candidates in different quality while SKT and PostKS assume there always exists GT-knowledge in their candidates. For REALM, the notification date of ICML 2020 is quite close to the submission date of NeurIPS 2020. A thorough discussion about these three works will be presented in the final version.

**For Reviewer #2:** We will follow your suggestions on the improvement of clarity in the final version. We have tried increasing the search space of knowledge with details described in **For Reviewer #1:**3(i). Increasing the search space will cost more GPU memory and training time, and thus it is not easy to scale.

**For Reviewer #3:** During test time, the knowledge candidates are already prepared in test dataset, we only need to tailor the knowledge candidates to meet the capacity constraint of UniLM by using knowledge selector $p(y = 1|C, Z)$.

**For Reviewer #4:** We will follow your suggestions on the improvement of clarity in the final version. "Zero-Resource" in our paper specifically refers to training model without crowd-sourced knowledge-grounded dialogues (e.g. Wizard, Topical Chat) rather than completely without any knowledge resources (e.g. Wikipedia). Existing knowledge-grounded dialogue generation models are all trained on crowd-sourced knowledge-grounded dialogue datasets. Such datasets are enormously expensive to collect, and thus are small in size (Wizard contains only about 150k utterances, for example). We explore a way to train such a model from easy-to-collect datasets (e.g. Reddit and Wikipedia) in the paper, so our training data could have up to about 2600k utterances. The generalization ability of our model originates from the ability of our proposed method to break through the limitation of training data size.

[Meta-Review · NeurIPS 2020]

This paper proposes a novel approach based on graphical models for zero-resource generation of knowledge-grounded dialogues. The approach introduces two latent variables to represent knowledge to ground the response on and to represent the degree of grounding, and is trained with variational inference. The proposed approach achieves comparable results to state-of the art methods trained with expensive-to-collect annotated data, without any such collection, and generalizes to unseen topics better than the supervised approaches. While a majority of reviewer questions are answered by the rebuttal, the reviewers still have a few points that could be clarified in the next revision.